# Systems approach for congruence and selection of cancer models towards precision medicine

Jian Zou[1], Osama Shah[2,3,4], Yu-Chiao Chiu[5,6], Tianzhou Ma[7], Jennifer M. Atkinson[2,3,4], Steffi Oesterreich[2,3,4]*, Adrian V. Lee[2,3,4]*, George C. Tseng[8]*

1 Department of Statistics, School of Public Health, Chongqing Medical University, Chongqing, China, 2 Women's Cancer Research Center, UPMC Hillman Cancer Center (HCC), Pittsburgh, Pennsylvania, United States of America, 3 Magee-Womens Research Institute, Pittsburgh, Pennsylvania, United States of America, 4 Department of Pharmacology & Chemical Biology, University of Pittsburgh, Pittsburgh, Pennsylvania, United States of America, 5 Cancer Therapeutics Program, UPMC Hillman Cancer Center (HCC), Pittsburgh, Pennsylvania, United States of America, 6 Department of Medicine, University of Pittsburgh, Pittsburgh, Pennsylvania, United States of America, 7 Department of Epidemiology and Biostatistics, University of Maryland, College Park, Maryland, United States of America, 8 Department of Biostatistics, University of Pittsburgh, Pittsburgh, Pennsylvania, United States of America

* oesterreichs@upmc.edu (SO); leeav@upmc.edu (AVL); ctseng@pitt.edu (GCT)

**Data Availability Statement:** An R package, CASCAM, with an interactive app is publicly available (https://github.com/jianzou75/CASCAM.) to facilitate the use of our proposed framework.

## Abstract

Cancer models are instrumental as a substitute for human studies and to expedite basic, translational, and clinical cancer research. For a given cancer type, a wide selection of models, such as cell lines, patient-derived xenografts, organoids and genetically modified murine models, are often available to researchers. However, how to quantify their congruence to human tumors and to select the most appropriate cancer model is a largely unsolved issue. Here, we present Congruence Analysis and Selection of CAncer Models (CASCAM), a statistical and machine learning framework for authenticating and selecting the most representative cancer models in a pathway-specific manner using transcriptomic data. CASCAM provides harmonization between human tumor and cancer model omics data, systematic congruence quantification, and pathway-based topological visualization to determine the most appropriate cancer model selection. The systems approach is presented using invasive lobular breast carcinoma (ILC) subtype and suggesting CAMA1 followed by UACC3133 as the most representative cell lines for ILC research. Two additional case studies for triple negative breast cancer (TNBC) and patient-derived xenograft/organoid (PDX/PDO) are further investigated. CASCAM is generalizable to any cancer subtype and will authenticate cancer models for faithful non-human preclinical research towards precision medicine.

## Author summary

Cancer research relies on models, such as cell lines, patient-derived xenografts (PDX), and patient-derived organoids (PDO), as essential alternatives to human studies.

**Funding:** Research funding for this project was provided in part by Susan G. Komen Scholar awards (SAC110021 to AVL and SAC160073 to SO), the Breast Cancer Research Foundation (to AVL and SO), the Magee Foundation, and the National Cancer Institute (CA252378). JZ and GCT were funded by NIH grant R01LM014142 and R21LM012752. This research was supported in part by the University of Pittsburgh Center for Research Computing, RRID:SCR_022735, through the resources provided. Specifically, this work used the HTC cluster, which is supported by NIH award number S10OD028483. The study is in part funded by P30CA047904. The funders had no role in study design, data collection and analysis, decision to publish, or preparation of the manuscript.

**Competing interests:** The authors have declared that no competing interests exist.

However, it is crucial to determine how well these models mimic human patients and to quantify their congruence in disease-relevant genes and regulatory pathways. As the number of cancer models grows, researchers face the challenge of selecting the most representative model based on molecular profiles. Existing methods are machine learning based and are limited to prediction using genome-wide information without mechanistic insights. To address this, we developed a comprehensive suite of bioinformatics tools, namely Congruence Analysis and Selection of CAncer Models (CASCAM). The framework develops a multi-stage systems approach to quantify pathway and gene specific congruence and allow prioritization and selection of the most congruent cancer model(s), which provides a paradigm shift towards gene regulatory and systems investigations.

## Introduction

Cancer models, inheriting genetic properties of the tumors of origin, are essential tools in cancer research for exploring carcinogenesis and developing drugs in basic, translational and clinical studies. For a given cancer subtype, a wide selection of models, such as cell lines, patient-derived xenografts (PDX), patient-derived organoids (PDO), and genetically modified murine models, are often available to researchers. Specifically, patient-derived cancer models, such as PDO, are increasingly available with molecular profiling and are expected to play a heightened role in disease understanding, drug response prediction and precision medicine [1, 2]. For example, the PDCM Finder (Patient Derived Cancer Model Finder), funded by National Cancer Institute, provides an open catalog of patient-derived cancer models with an established "minimal information standard" for researchers to upload new cancer models [3], which currently includes 4,661 xenograft models, 1547 cell lines and 108 PDO as of 10/20/2022.

Despite advances in technology and reduced cost, cancer models can be mislabeled [4] and genomic/epigenomic alterations may accumulate across passages in culture. Many cancer models may be potentially mis-annotated from their origins or the quality of congruence may vary or decay over time [4–6]. Due to increasing availability of new cancer models and associated comprehensive omics data, evaluation and comparison of cancer models with human tumors using transcriptomic and multi-omics data have drawn increasing attention in recent years [4, 5, 7–18]. However, existing evaluation tools mostly belong to two major categories, congruence (correlation-based) analysis and authentication (machine-learning-based) analysis, and do not sufficiently serve the purpose of identifying appropriate models for precision medicine. In congruence analysis, correlation/association measures are usually applied to quantify similarity of a cancer model to the target tumor cohort in a genome-wide scale [5, 10, 11, 14, 15, 18]. In contrast, authentication analysis develops machine learning models, such as suitability score [16], random forest, ridge regression and nearest template prediction, for accurate assignment of cancer models to human cancer types. S1 Table outlines features and shortcomings of the existing methods. Overall, these tools have significant limitations in the following four areas: (1) Machine-learning-based authentication methods focus on predication accuracy but are not designed to prioritize candidate cancer models that best mimic the target tumor cohort; (2) On the other hand, correlation-based congruence methods can prioritize cancer models but they often produce lower prediction accuracy; (3) Current congruence or authentication methods cannot characterize pathways or molecular mechanisms that are most or least mimicked by a cancer model, which is essential in precision medicine development; (4) Data compatibility and harmonization between cancer model and human tumor data have

not been systematically considered and evaluated in the current literature, which is a critical step to achieve high accuracy and avoid misleading mechanistic conclusions.

To this end, we developed CASCAM with three modules to overcome the aforementioned shortcomings of existing methods (see Fig 1). In the first "data harmonization" module, we applied the recently developed Celligner method to correct for batch effects and obvious variations between cancer models and tumors that prevent analysis of congruence. In the second "transparent machine learning pre-selection" module, we developed a transparent machine learning approach, integrating prediction assignment probability from sparse linear discriminant analysis (SDA) and deviance score derived from the SDA projected space. The integrative framework combines advantages of high classification accuracy by machine-learning-based authentication analysis and prioritization by correlation-based congruence analysis to pre-select, say, the top $\sim 10$ promising cancer models from up to hundreds of initial candidates. The pre-selected cancer models then enter the final "pathway and mechanistic-based selection" module. By integrating pathway and regulatory network information, multiple bioinformatic and visualization tools, including differential expression, pathway enrichment analysis, heatmaps, violin plots and topological network plots, iteratively investigate disease-relevant biological mechanisms that are best or least mimicked by each cancer model. We note that the two-stage selection by global (genome-wide congruence) pre-selection in Module 2 and then targeted (pathway- and gene-based congruence) evaluation in Module 3 is an essential and innovative aspect of CASCAM. We demonstrate that the highest genome-wide congruent cancer models selected from Module 2 may not harbor critical pathways and genes relevant to the target tumor subtype and thus show a lower score in essential pathways. On the other hand, pre-selection in Module 2 is necessary to reduce the number of cancer model candidates for allowing detailed mechanistic investigation in Module 3.

For demonstration purposes, two case studies in this paper focused on invasive lobular breast carcinoma (ILC), a histological subtype containing 10–15% of all breast cancers and

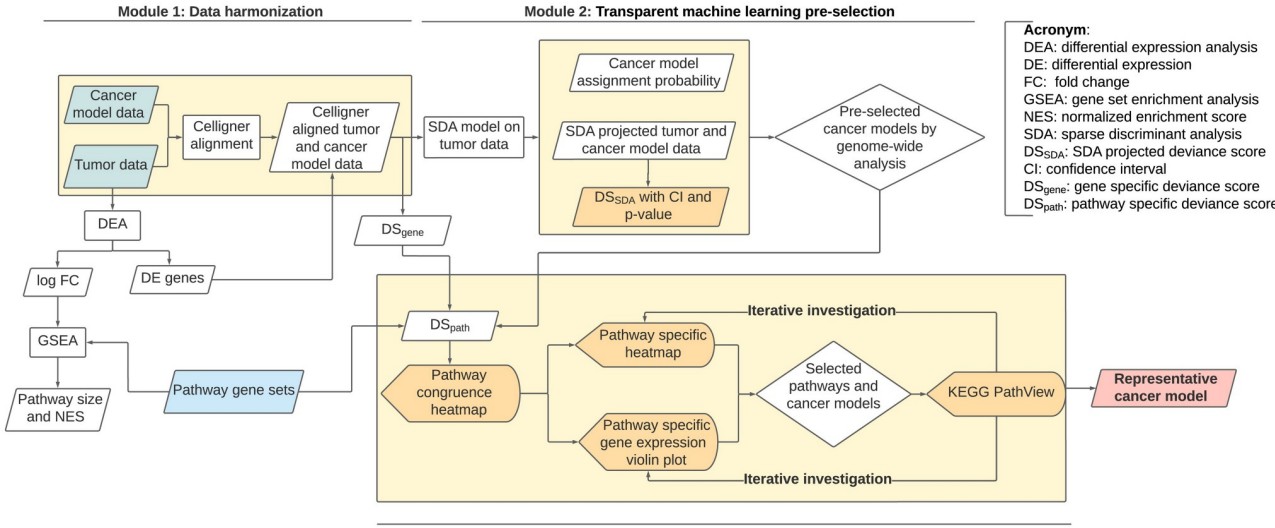

**Fig 1. Flowchart of CASCAM for congruence quantification and selection.** Tumor and cancer model gene expression data are first harmonized (Module 1). Transparent machine learning by sparse discriminant analysis (SDA) is applied by combining predication accuracy and SDA-based deviance score for pre-selecting candidate cancer models (Module 2). Pathway-specific mechanistic explorations are iteratively investigated to conclude the final representative cancer model (Module 3). Blue frames represent input data, orange frames for essential output results, parallelogram frames for intermediate results, rectangular frames for analysis process, bullet-shaped frames for visualization, and rhombus frames for decision making.

with a hallmark genomic feature consisting of *CDH*1 gene (E-cadherin) mutation and subsequent loss of cell-cell adherent junctions. There is a compelling need to develop and identify representative cancer models for ILC since previous breast cancer models mostly focus on the more prevalent ($\sim 80\%$) invasive ductal carcinoma (IDC) subtype (also known as no special type (NST)). Indeed, there are very few ILC annotated cell lines publicly available; however, a previous study identified numerous breast cancer (BC) cell lines which lack ILC annotation but harbor *CDH*1 mutations—they were named 'ILC-like' and these potentially could serve as representative models of human-ILC disease [19]. Beyond ILC, we performed an additional analysis on identifying the most representative cell line for triple negative breast cancer (TNBC) and note that CASCAM is applicable in general cancer research by quantifying congruence and identifying the most appropriate cancer model for any given tumor (sub)type.

## Results

### Case study 1: Selection of cell line for ILC

**Data harmonization between cancer model and tumor transcriptomic data.**   The critical first step for quantifying congruence and selection of cancer model(s) is to ensure omics data harmonization between cancer models and human tumors. This is important as cell lines do not contain many genes expressed in the tumor microenvironment. We accessed bulk transcriptomic data of 9,264 pan-cancer tumor samples across 24 cancer types from The Cancer Genome Atlas Program (TCGA) (960 samples are breast cancer, BC), and 1,257 pan-cancer cell lines from Cancer Cell Line Encyclopedia (CCLE) and Invasive Lobular Cancer Cell Line Encyclopedia (ICLE) (65 annotated as BC cell lines) (see Materials and methods section). We then evaluated performance of normalization using five approaches—A) no normalization; B) quantile normalization [20] to normalize BC tumors and BC cell lines; C) ComBat [21] to normalize BC tumors and BC cell lines; D) Celligner [5] to normalize BC tumors and BC cell lines; E) Celligner to normalize pan-cancer tumors and pan-cancer cell lines. We visualized the relationship of BC tumors and BC cell lines when different normalization approaches were applied using UMAP [22] (see in Fig 2). Biased separation of tumors and cell lines was clearly found when no normalization or conventional quantile normalization were implemented (see Fig 2A and 2B). Combat and Celligner using BC tumors and BC cell lines (see Fig 2C and 2D) produced improved normalization although systematic bias was still observed from small clusters of cell lines, showing insufficient quality of data harmonization. In contrast, Celligner using pan-cancer tumors and pan-cancer cell lines (see Fig 2E) best eliminated batch effects between BC tumors and BC cell lines.

To further examine the quality of Celligner normalization in approach E (Fig 2E), we investigated eight cell lines each with three experimental replicates from different sources (see Materials and methods section) and confirmed their high reproducibility in the UMAP plot (Fig 3A). From breast cancer subtype annotation [23], we confirmed that TCGA tumors in the lower-right cluster were mostly annotated as basal-like (118 out of 160) (Fig 3B). Reassuringly, 26 of 28 CCLE cell lines in that cluster were also annotated as basal-like. As a result, we performed Celligner normalization before all down-stream analysis in this paper.

**Transparent machine learning pre-selection.**   We next extended and applied a transparent machine learning (ML) method, namely sparse discriminant analysis (SDA), and combined with a deviance score (DS) derived from SDA to pre-select from up to hundreds of candidate cancer models and narrow down to <10 of the most promising cancer models. The machine learning (ML) setting here was similar to existing literature, where a prediction model was constructed using human tumor data (e.g., TCGA) as the training set and then was used to classify cancer models to the targeted group (ILC) versus comparison group (IDC). To

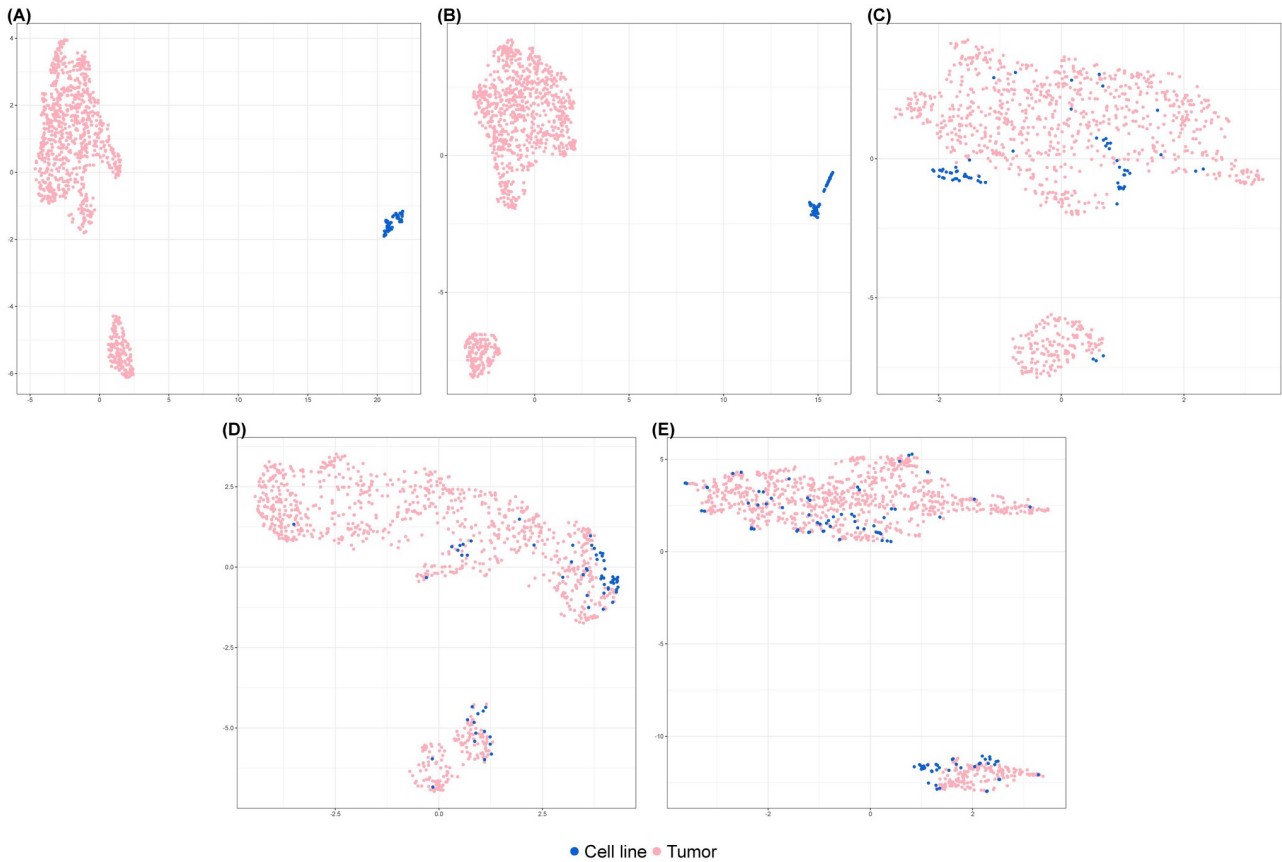

**Fig 2. UMAP for comparison of multiple data harmonization approaches.** UMAP for normalized BC tumors (n = 960) and BC cell lines (n = 65) to compare five normalization approaches: (A) no correction and (B) quantile normalization (C) ComBat and (D) Celligner utilizing BC tumors and BC cell lines (E) Celligner utilizing pan-cancer tumors and pan-cancer cell lines. The final approach best eliminates batch effects by mixing well the BC tumors and BC cell lines.

justify application of SDA, Table 1 shows performance of 16 popular machine learning methods, six of which were used to classify cancer models according to TCGA cancer types in the literature. Detailed description of these machine learning methods can be found in S1 Table and Materials and methods section. In existing publications, machine learning analysis aimed to classify cancer models into major cancer types, such as the 24 cancer types in TCGA. We note that since the two subtypes we focus on (ILC and IDC) are two histological subtypes within breast cancer, the differences are more subtle. The machine learning and congruence analysis tasks are expected to be more difficult but biologically more impactful.

Table 1 shows evaluation result of the 16 machine learning methods in BC machine learning tasks from three different aspects (tumor type, histological subtype, and molecular subtype). Convolutional neural network (CNN) is a category of deep learning methods commonly designed for classification problems [24–26]. In this study, we included three CNN models initially optimized for pan-cancer classification [25]. Columns 2–4 contain prediction accuracy results: 5-fold cross validation of ILC versus IDC using TCGA BC data, ER+ versus ER- classification using TCGA as training data and CCLE as test data, and BC versus other cancer types using TCGA as training data and CCLE as test data. The result shows SDA and elastic net to have the highest average accuracy, followed by 2D-Hybrid-CNN and ridge regression methods. Specifically, SDA achieved 91% accuracy for ILC vs IDC cross-validated tumor

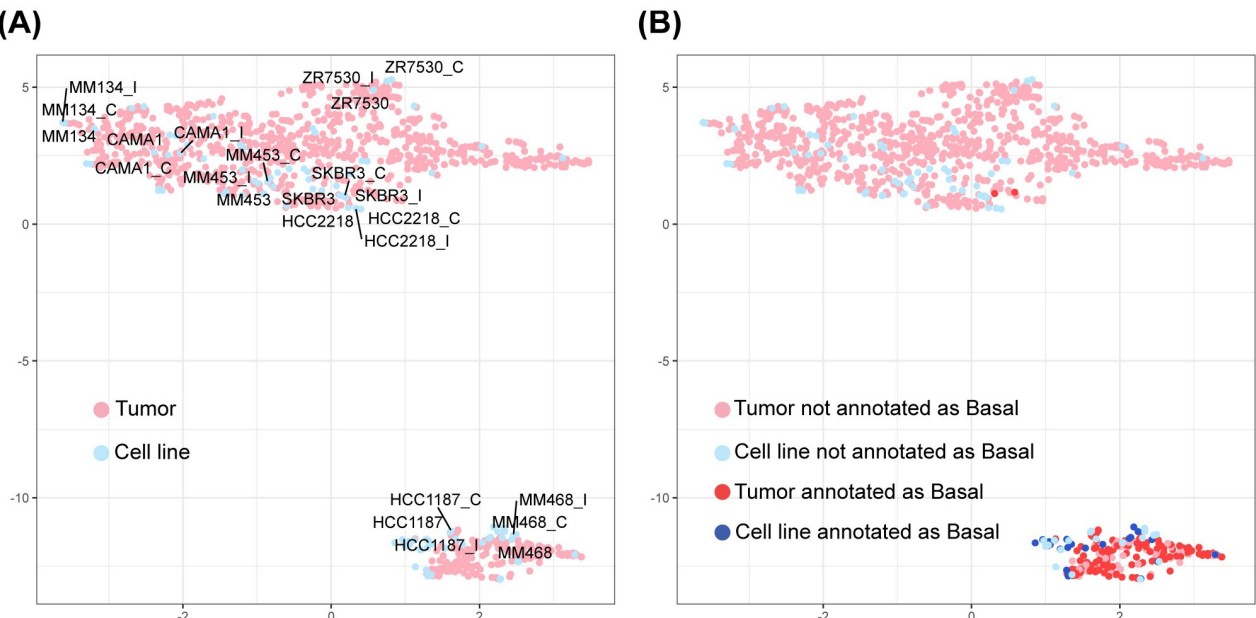

**Fig 3. UMAP after data harmonization with replicates and basal subtype information.** (A) Three replicates (cell line; cell line_C; cell line_I) for each of the eight cell lines are highly reproducible. (B) The lower-right cluster contains dominantly tumors and cell lines annotated as basal-like (118/160 tumors and 26/28 cell lines).

**Table 1. Evaluation and properties of 13 popular machine learning methods.** Six methods applied for cancer model prediction in previous papers are highlighted (*). Prediction accuracies are shown in three machine learning evaluation examples. Parentheses in the second column are standard deviations of accuracies in five repeats of five-fold cross-validation.

| | Machine learning evaluation | | | Machine learning relevant properties | | |
|---|---|---|---|---|---|---|
| | ILC vs IDC | ER+ vs ER- | BRCA vs other cancers | Gene selection | Assignment probability | Deviance score |
| | TCGA; 5-fold CV | Training data: TCGA; Test data: CCLE | Training data: TCGA; Test data: CCLE | | | |
| SDA | 0.91 (0.02) | 0.91 | 0.86 | Yes | Yes | Yes |
| ElasticNet | 0.90 (0.03) | 0.93 | 0.85 | Yes | Yes | No |
| 2D-Hybrid-CNN | 0.87 (0.03) | 0.93 | 0.86 | No | No | No |
| RidgeRegress* | 0.88 (0.02) | 0.91 | 0.84 | Yes | Yes | No |
| Pearson25* | 0.86 (0.01) | 0.86 | 0.9 | No | No | No |
| KNN | 0.85 (0.03) | 0.86 | 0.91 | No | Yes | No |
| 2D-Vanilla-CNN | 0.86 (0.04) | 0.88 | 0.85 | No | No | No |
| 1D-CNN | 0.86 (0.03) | 0.86 | 0.86 | No | No | No |
| RandomForest* | 0.85 (0.01) | 0.91 | 0.82 | Yes | Yes | No |
| RSLDA | 0.81 (0.11) | 0.77 | 0.86 | Yes | Yes | Yes |
| CancerCellNet* | 0.79 (0.03) | 0.82 | 0.79 | Yes | Yes | No |
| LDA | 0.80 (0.03) | 0.68 | 0.82 | No | Yes | Yes |
| NTP | 0.61 (0.03) | 0.86 | 0.82 | No | No | Yes |
| SpearmanMed* | 0.40 (0.03) | 0.84 | 0.61 | No | No | Yes |
| PearsonMed* | 0.38 (0.04) | 0.84 | 0.62 | No | No | Yes |
| Logistic | 0.52 (0.04) | 0.43 | 0.65 | No | Yes | No |

classification, 91% for ER+ vs ER- cell line prediction and 86% for BRCA vs other cancers in cell line prediction. The CNN methods produced reasonably high accuracy in the three tasks but not among the best, possibly due to limited sample size. 2D-Hybrid-CNN was proposed to benefit from having two-dimensional inputs with simple one-dimensional convolution operations and had better performance than 1D-CNN and 2D-Vanilla-CNN, consistent with previous results in the cancer subtype classification [25].

In addition to binary prediction accuracy performance, Table 1 lists three machine-learning relevant properties that are critical for evaluation and selection towards precision medicine: feature (gene) selection, prediction assignment probability and deviance measure. Explicit gene selection identifies gene signatures involved in the prediction model and provides transparent machine learning. Assignment (prediction) probability reports prediction confidence and ranking for cancer models predicted into the target tumor subtype. Many methods, including ElasticNet and 2D-Hybrid-CNN, produce hard label predictions, while only a few, such as LDA, SDA, and RSLDA, offer soft assignments with classification probabilities. However, these soft assignment probabilities often tend towards binary values, as evidenced in S1 Fig. This limits their utility for model prioritization and underscores the limitation of relying solely on classification probabilities for model prioritization. Consequently, we devised a strategy that combines classification probabilities with deviance scores to more effectively prioritize and pre-screen cancer models, as detailed in Module 2. Finally, deviance measure (e.g., dissimilarity measure or lack-of-association measure) provides supplemental information to prediction assignment probability for cancer model suitability.

Of the 16 methods in Table 1, only SDA and a robust variant, RSDA, can be extended for all three transparent machine learning properties. Taken together, SDA was among the most accurate machine learning methods and provided three essential properties of gene selection, assignment probability (denoted as $P_{SDA}$) and deviance score (denoted as $DS_{SDA}$); it was chosen to be the core machine learning method in CASCAM. Particularly, we defined deviance score $DS_{SDA}$ as the (signed) standardized distance between a cell line to the center of the target tumor cohort on the SDA projected space. Bootstrap analysis was then performed in the tumor data to calculate the confidence interval and two-sided p-value, denoted as $pval(DS_{SDA})$ (see Materials and methods section for details).

Since nearly all ILC cases are luminal ER-positive (i.e. basal negative or non-basal [27]), we focused on the large luminal non-basal cluster in Fig 3B, which contains 798 BC tumors and 37 BC cell lines. DU4475 [28] was manually included to explore the performance of a basal positive cell line. Of the 38 cell lines, we pre-selected 14 candidate cell lines using Module 2 by high prediction assignment probability ($P_{SDA} > 0.5$) and small deviance score such that the corresponding p-value is not statistically significant (i.e., $pval(DS_{SDA}) > 0.05$) (S3 Table). We note that the selection of cell lines with large *p*-value is not conventional since hypothesis testing with small *p*-value is usually designed for capturing differences, while we are interested in congruence (i.e., no difference) here. S2 Table contains the 153 genes selected by SDA for constructing machine learning model: Column 1 represents gene weights for the model and the other columns are harmonized gene expression for all evaluated cell lines, sorted by $DS_{SDA}$. The pathognomonic feature of ILC is mutation of *CDH*1 and a subsequent reduction *CDH*1 mRNA expression. Thus, as expected the weight for *CDH*1 was -10.428 and was at least 5–10 fold greater than all the other predictive genes.

The necessity of using $P_{SDA}$ and $DS_{SDA}$ simultaneously can be seen in the SDA projected scatter plot of the 38 cell lines in Fig 4A. If we only used $P_{SDA}$ information (marked by red color), cell lines such as UACC812 and ZR751 were predicted to be ILC with $P_{SDA} = 100\%$, disregarding the fact that these cell lines' expression patterns were highly dissimilar to the averaged expression pattern (center) of ILC tumor cohort on SDA projection (large $DS_{SDA} = 2.991$

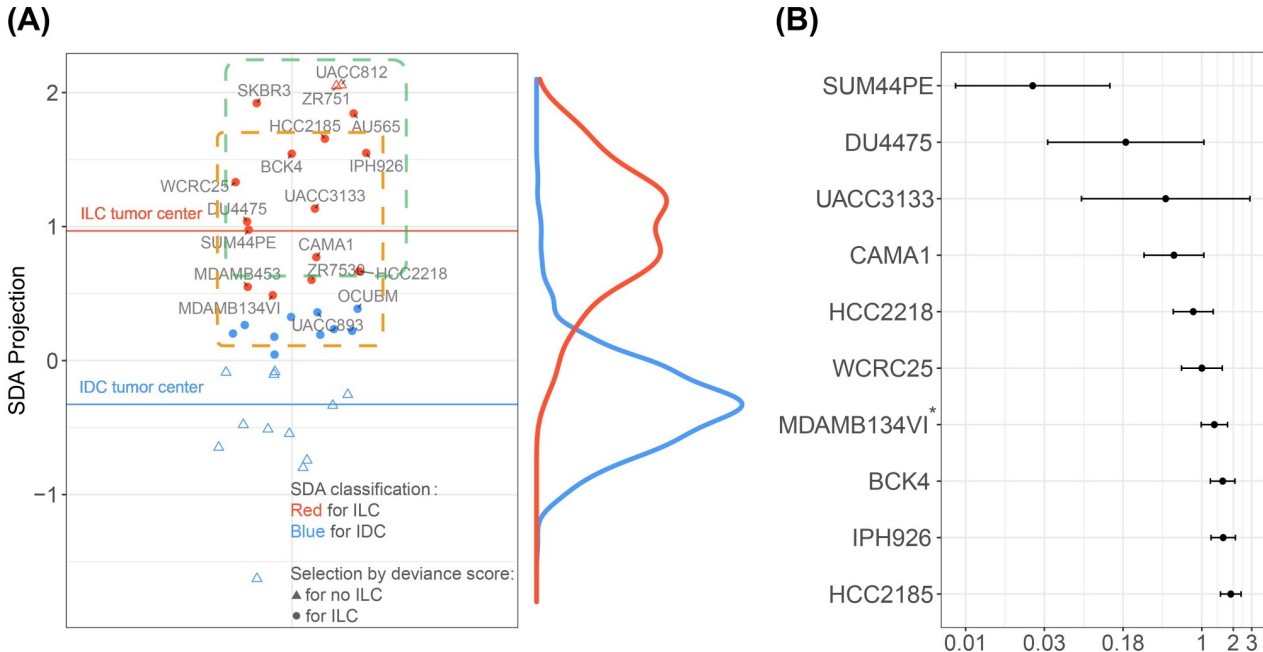

**Fig 4. Genome-wide cell line congruence and pre-selection.** (A) SDA projected scatter plot. y-axis represents the projected values for 38 cell lines, the red and blue horizontal lines represent the median projected value (center) of ILC and IDC tumor samples respectively. The density plots on the right shows distributions of 769 IDC (blue) and 191 ILC (red) tumors. Red color of the dots represents SDA classification to ILC (threshold $P_{SDA} >$ 50%), and the solid dots represent small SDA-based deviance scores (threshold $pval(DS_{SDA})>0.05$). More stringent criteria were indicated by the dashed rectangle. Cell lines with $P_{SDA} > 0.8$ were enclosed by green dashed rectangle and the ones with $pval(DS_{SDA})>0.1$ were enclosed by orange dashed rectangle. (B) SDA projected deviance score (absolute value) with 95% confidence interval. 9 unbiased-selected and 1 manually-included (marked with asterisk) cell lines are ranked based on $|DS_{SDA}|$, and 95% confidence intervals are obtained by bootstrap analysis on log-scale.

and 2.970 to ILC, respectively) (S3 Table). In contrast, if we only used $DS_{SDA}$ (marked by round shape), cell lines such as OCUBM and UACC893 had a relatively small deviance score to ILC ($DS_{SDA} = 1.597$ and $1.667$ to ILC, respectively), but they were also close to the center of IDC tumor cohort. By applying the combined criteria of $DS_{SDA}$ and $P_{SDA}$, 14 of the 38 cell lines were identified as well-resemblance to the ILC subtype ($P_{SDA} = 54.8 - 100\%$ and $DS_{SDA} = 0.024-1.892$). Specifically, CASCAM identified SUM44PE ($DS_{SDA} = 0.024$ and $P_{SDA} = 98.7\%$) and DU4475 ($DS_{SDA} = 0.188$ and $P_{SDA} = 99.2\%$) as the two most genome-wide representative cell lines for ILC. UACC3133 was ranked the third with small deviance $DS_{SDA} = 0.452$ but had a wide 95% confidence interval [0.067, 3.040]. The congruent finding of SUM44PE is consistent with literature, as it has been reported to have anchorage-independence and limited migration and invasion ability, which are unique properties to the ILC-like cell lines [29] and are widely studied in ILC [30, 31]. To avoid the ambiguous assignment probabilities, we further restricted the selection criteria to $P_{SDA} > 0.8$ (enclosed by green dashed rectangle) and $pval(DS_{SDA})>0.1$ (enclosed by orange dashed rectangle). ZR7530 ($P_{SDA} = 0.764$), MDAMB453 ($P_{SDA} = 0.670$), SKBR3 ($DS_{SDA} = 2.615$, $pval = 0.052$) and AU565 ($DS_{SDA} = 2.405$, $pval = 0.062$) were filtered out, and the 9 cell lines that met the criteria were used for further investigation. We note that MDA-MB-134VI ($P_{SDA} = 0.548$) was manually included for further evaluation as it is widely used in ILC research [32, 33] In Fig 4B, the 9 unbiased-selected and 1 manually-included cell lines were ranked by $DS_{SDA}$ with 95% confidence interval provided, with SUM44PE and HCC2185 occupying the highest and lowest ranks, respectively.

**Pathway and mechanistic-based selection of cancer model(s).** Next, we applied Module 3 with pathway-specific and gene-specific evaluation for further prioritization of the 10 pre-

selected breast cancer cell lines. While SDA and other machine learning methods offer gene selection aimed at maximizing predictive accuracy, they can miss driver or regulatory genes essential for mechanistic understanding [34]. Therefore, in our CASCAM framework, machine learning via SDA serves only as a pre-screening tool in Module 2. The subsequent gene-based and pathway-based deviance scores, along with mechanistic investigations in Module 3, are crucial for conducting biologically insightful congruence analysis.

Using a similar definition of $DS_{SDA}$, we calculated gene- and pathway-specific deviance scores, $DS_{gene}$ and $DS_{path}$, for characterizing congruence of each candidate cell line. Differential expression analysis on 769 IDC versus 191 ILC samples in TCGA identified 3,065 DE genes. For pathway investigation, 236 pathways in Hallmark and KEGG from MSigDB [35] were first identified, and 53 pathways with more than 20 DE genes were used for GSEA pathway analysis [36].

We visualized pathway-specific deviance scores ($DS_{path}$) for the 53 selected pathways (rows) and 10 selected cell lines (columns) in the heatmap (see in S2 Fig). The side-bar on the top shows genome-wide congruence $DS_{SDA}$ for each cell line and the side-bar on the left margin shows size and normalized enrichment score (NES) for each pathway. In general, genome-wide resemblance does not guarantee similar performance in specific pathways. SUM44PE, for example, was the most congruent ILC cell line with the smallest $DS_{SDA}$. However, it was second to worst congruent cell line in Hallmark heme metabolism, where heme is an iron-containing porphyrin with multifaceted roles in cancer ($DS_{path}$ = 1.029). When users have prior knowledge of known relevant pathways, the most congruent cell line can be selected by the smallest averaged $DS_{path}$ of the pre-selected pathways. If no prior biological knowledge is used, we recommend using pathways with adequate pathway size (e.g., 30< size <200) and enrichment (e.g., |$NES$| >1.5) for final cell line decision. This criterion selected 14 pathways and the heatmap of their pathway-specific deviance score was shown in Fig 5A (detailed values available in S4 Table). Among the 14 pathways, the majority of pathways were cancer related (marked star in Fig 5A). For example, Hallmark E2F Targets has the most significant NES ($NES$ = −2.18, adjusted $pval$ < 0.0001, 79 DE genes), which includes genes encoding cell cycle related targets of E2F transcription factors. Related to the loss of E-cadherin, E2F was reported to show difference in ILC compared with IDC [30, 37]. KEGG PPAR Signaling Pathway, including genes related to peroxisome proliferator-activated receptors (PPARs) signaling, is significantly enriched ($NES$ = 1.55, adjusted $pval$ = 0.048, 22 DE genes), and is also widely reported for its upregulation in ILC tumors in multi-omics studies [38, 39].

Given that loss of $CDH$1 [40] and subsequent dysfunction of cell-cell adhesion [41] is the hallmark of ILC, we manually included "KEGG Cell Adhesion Molecules" pathway for analysis shown in Fig 5 and S4 Table, in addition to the 14 unbiased selected pathways. The pathway was not selected because its |$NES$| = 0.854 did not meet the prespecified criterion. The second to the last row in criterion. The second to the last row in Fig 5A shows average $DS_{path}$ of the 14 pathways for each cell line, in which CAMA1 had the smallest average deviation. CAMA1 was also congruent to ILC in the "KEGG Cell Adhesion Molecules" pathway (22 DE genes, $DS_{path}$ = 0.468, $pval$ = 0.634). Although SUM44PE, DU4475 and UACC3133 outperformed CAMA1 in the genome-wide SDA-based deviance score (Fig 4), each of them did not mimic well in at least part of the 14 pathways (one circle: $pval$ < 0.1; two concentric circles: $pval$ < 0.05; three concentric circles: $pval$ < 0.01, showing non-congruence) while CAMA1 had uniformly high congruence. For example, DU4475 did not mimic ILC in several important cancer and ILC-related pathways, such as "Hallmark TNFA Signaling Via NFKB", "Hallmark Glycolysis", "Hallmark MTORC1 Signaling", etc.

For a pathway of interest, CASCAM further generated a gene-specific deviance score ($DS_{gene}$) heatmap. The heatmap of 22 DE genes in the "KEGG Cell Adhesion Molecules"

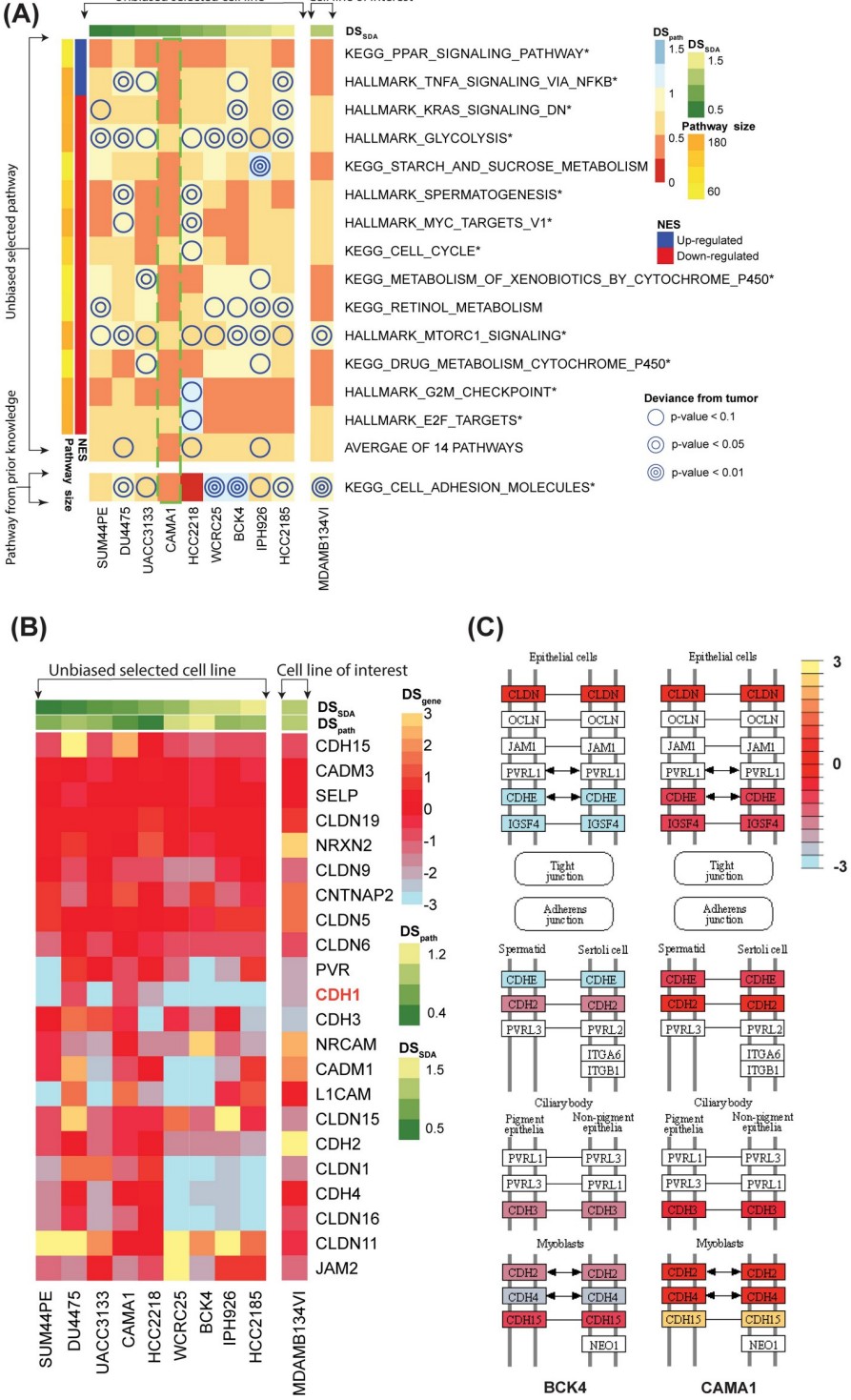

**Fig 5. Pathway- and gene-specific analysis for selection of representative cell line(s).** (A) Heatmap of pathway-specific deviance scores ($DS_{path}$) with 14 unbiased-selected and 1 manually-included pathways ($30 < size < 200$, $|NES| > 1.5$; shown on the rows) and 9 unbiased-selected and 1 manually-included cell lines (columns). The genome-wide SDA projected deviance score ($DS_{SDA}$) is shown on the top side-bar and the pathway size and normalized enrichment score (NES) are on the left. Positive (negative) NES indicates up-regulation (down-regulation) in ILC compared to IDC. Average of the 14 pathways and the pre-selected "KEGG Cell Adhesion Molecules" pathway are shown at the bottom. The p-values of $DS_{path}$ are annotated in the heatmap (one circle: $p-value < 0.1$; two concentric circles: $p-$

*value* $< 0.05$; three concentric circles: $p - value < 0.01$), and smaller p-values indicate worse congruence. (B) Gene-specific heatmap shows $DS_{gene}$ for the 10 selected cell lines and 22 DE genes in "KEGG Cell Adhesion Molecules" pathway. (C) Part of KEGG PathView topological networks for BCK4 ($DS_{path} = 1.323$) for the "KEGG Cell Adhesion Molecules" pathway. The result shows discordance of 10 genes in BCK4 (orange stars showing up-regulation compared to ILC tumors and blue start showing down-regulation).

pathway provides gene-level resolution of congruence information (Fig 5B). BCK4 appeared to be the least congruent cell line in this pathway with 10 genes having large deviance scores (|$DS_{gene}$|>2; Fig 5B). Next, we utilized KEGG topological regulatory network information [42] to investigate gene-specific congruence to ILC in KEGG Cell Adhesion Molecules for selected cell lines. The well-known ILC hallmark gene *CDH1* only showed congruence in CAMA1 and DU4475 (*CDH*1 highlighted in Fig 5B). Although the MDA-MB-134VI cell line has been widely used in ILC research, it was shown to have similar congruence to both ILC and IDC (Fig 5A and S3 Table, $P_{SDA}$ to ILC = 0.548). Furthermore, although MDA-MB-134VI was congruent to ILC in many of the 14 pathways, it was not congruent in the "KEGG Cell Adhesion Molecules" pathway and many ILC-relevant genes in this pathway (Fig 5A and 5B). Among the 10 cell lines, the BCK4 cell line ($DS_{path} = 1.323$, $pval = 0.005$) had the largest $DS_{path}$, indicating worst genome-wide congruence (Figs 5B and S3). We employed KEGG PathView plots to illustrate the gene network discrepancies in the Cell Adhesion Molecules pathway, focusing on BCK4 and CAMA1 (see Fig 5C for part of the network).

BCK4 had many discordant genes to ILC: 2 genes highly up-regulated to ILC ($DS_{gene} > 2$; *CLDN*11 and *NRCAM*) and 8 genes highly down-regulated to ILC ($DS_{gene} < -2$; *CADM*1, *CDH*1, *PVR*, *L1CAM*, *CLDN*1, *CLDN*16, *CDH*4, and *JAM*2). Of these genes, cadherin genes (*CDH*1, *CDH*4 and *CDH*15) were cell adhesion molecules that are critical in the formation of adhesion junctions for cells to adhere to each other [43]. Similarly, claudin genes (*CLDN*1, *CLDN*11 and *CLDN*16) are proteins essential for the formation of tight junctions in epithelial and endothelial cells [44].

As shown in this ILC representative cell line selection example, CASCAM provided multiple visualization tools and interactive software functions, including violin plot (S3 Fig), pathway-specific congruence heatmap ($DS_{path}$; Fig 5A), gene-specific congruence heatmap ($DS_{gene}$; Fig 5B), and KEGG topological network plot (Fig 5C), to allow researchers to iteratively investigate concordance and discordance of cell lines with the target tumor cohort. In conclusion, 5 of the 10 cell lines are determined as ILC-like in the "KEGG Cell Adhesion" pathway (*pval* ($DS_{path}$)>0.05)), and we recommend them as appropriate ILC cell lines in the order of average $DS_{path}$ of the 14 selected pathways: CAMA1 ($DS_{path} = 0.505$), UACC3133 ($DS_{path} = 0.667$), SUM44PE ($DS_{path} = 0.689$), HCC2218 ($DS_{path} = 0.748$), IPH926 ($DS_{path} = 0.754$).

**Specificity evaluation for irrelevant cell lines from other cancers.**   To evaluate the specificity of CASCAM when an irrelevant cell line belonging to neither ILC nor IDC is included, we selected 24 liver cancer cell lines from CCLE as negative controls in the analysis. As shown in S4(A) Fig, five of the 24 liver cell lines were selected under the same classification probability $P_{SDA}$ and deviance score $DS_{SDA}$ criteria. When further analyzed by the deviance score heatmap of selected pathways in S4(B) Fig, the five selected liver cell lines exhibit extremely large deviance scores in all pathways, demonstrating the increased specificity to avoid false positives by pathway-based mechanistic evaluation in Module 3 and the shortcoming of relying on Module 2 machine learning alone.

## Case study 2: Selection of cell line for TNBC

To demonstrate the generalizability of CASCAM beyond ILC samples, we applied the algorithm to three major molecular subtypes of breast cancer (HER2+, ER+/PR+/HER2- and

TNBC) [45] using the TCGA and CCLE datasets. By matching the samples with their molecular information [46, 47], our analysis included 326 patients with ER+/PR+/HER2-, 152 with HER2+, and 98 with TNBC, along with 6, 12, and 25 cell lines for these respective subtypes.

On the genome-wide analysis, we employed a one-vs-rest approach with the same threshold before and the result is shown in (S5 Table). CASCAM predicted 8 and 12 cell lines to HER2 + and TNBC, respectively, which is consistent to existing annotations. However, among the seven cell lines categorized to ER+/PR+/HER2-, two cell lines showed discrepancy (HCC1500 originally annotated to TNBC and UACC812 originally annotated to HER2+). Notably, HCC1500's annotation has varied in the literature, oscillating between ER+/HER2- and ER-/HER2- [48–52]. This heterogeneity might explain the false prediction of this cell line. Overall, CASCAM achieves 25/27 = 92.6% agreement between its prediction and the existing annotation.

Among the 12 cell lines identified as TNBC on the genomic level, we further narrowed the analysis to WNT signaling pathway to identify the most representative cell lines. The WNT signaling pathway plays an important role in TNBC, affecting various cellular processes such as metastasis, cell proliferation and apoptosis [53–55]. As shown in S5 Fig, HCC1599 and BT549 were found to be the best and the worst congruent model, respectively, in terms of WNT signaling. Thus, CASCAM is useful in and suitable for identifying biologically relevant cell line models outside of ILC and in other cancer types.

## Case study 3: Selection of PDO and PDX for ILC

To extend the algorithm to PDO and PDX, we applied CASCAM to 11 PDO and 136 PDX breast cancer models from the NCI PDMR (https://pdmr.cancer.gov/) database to select congruent cancer models for ILC versus IDC. These 147 cancer models were first normalized by the data harmonization module with the 9,264 TCGA pan-cancer tumor samples, and 960 TCGA BC samples were used for further investigation after normalization. UMAP in S6(A) Fig showed three distinct clusters. Except for the basal and non-basal group observed before (Fig 2), there was a small third cluster with 15 samples (2 PDO models, 12 PDX models, and 1 TCGA sample) from four patients. All four samples were annotated with triple-negative IDC, and two patients from PDMR have a metaplastic squamous cell carcinoma diagnosis. Due to the rare and unique subtype features of these tumors, we excluded these samples from further analysis and reproduced UMAP in S6(B) Fig. Similarly, we also excluded the samples in basal cluster, and 4 PDO and 25 PDX models were then kept for downstream analysis. The normalization result demonstrated excellent performance of Celligner for PDO and PDX.

We next applied the criteria ($P_{SDA} > 0.5$ and $pval(DS_{SDA}) > 0.05$) in the "transparent machine learning pre-selection" module and identified four candidate cancer models (3 PDX and 1 PDO) to represent ILC tumors (S6 Table). Cross-referencing with the PDMR database revealed that all four cancer models originated from the same patient (PRMR ID:171881–019-R). Table 2 showed 5 PDX and 1 PDO (denoted as PDO.1) originate from this patient. The 5 PDX samples contained one sample with passage 0 (denoted as PDX.0), two samples with passage 1 (denoted as PDX.1A and PDX.1B), and two samples with passage 2 (denoted as PDX.2A and PDX.2B). Intriguingly, the three highly congruent ILC PDX models were of passage 0 and 1 (PDX.0, PDX.1A and PDX.1B) while two PDX models with passage 2 (PDX.2A and PDX.2B) were not selected. A clear pattern emerged in Fig 6A, indicating that PDX.0 exhibits nearly perfect $DS_{SDA}$ congruence to represent ILC but the deviance score increased with increasing passage numbers (also see Table 2 Column 5), suggesting that the xenografts may evolve and be affected by the microenvironments in mice and deviate from the original tumor over time.

Stromal cells, such as fibroblasts and immune cells, have been reported to diminish across PDX passages [56]. To assess their role in passage variability (Fig 6A), we queried the Molecular Signatures Database (MSigDB) with "fibroblast AND immune" for all pathways and "fibroblast OR immune" for hallmark (H) and curated (C2) gene sets [57, 58]. This yielded 108 pathways, of which 7 were manually selected for their relevance to immune and fibroblast functions, comprising 357 genes. Among these, 35 were differentially expressed (Fisher's exact test, *pval* = 0.053). Analysis across six cancer models showed a weak correlation between $DS_{SDA}$ and a derived score, $DS_{path}$, based on these 357 genes (Pearson correlation = 0.16, *pval* = 0.76, S9(A) Fig). Conversely, pathways such as xenobiotic metabolism by cytochrome P450 (KEGG ID: hsa00980) and cell death (MSigDB ID: M5902) are strongly correlated with $DS_{SDA}$ (Pearson correlation = 0.91 and 0.87; *pval* = 0.01 and 0.02, respectively, S9(B)–S9(C) Fig). These results seem to indicate a minor role for immune and fibroblast functions in PDX passage variability. This conclusion is, however, suggestive and should be interpreted with caution due to the small number of cancer models (N = 6) and the use of bulk RNA analysis. For a more solid conclusion, future studies should employ higher-resolution methods such as scRNA-seq.

We next investigated clinical annotation of the patient PRMR ID:171881–019-R. Although the specific histological subtype was not annotated in the database, an insertion frameshift mutation (p.T115Nfs*53) for *CDH*1 was reported. Since loss of *CDH*1 is a key determinant of ILC, it strongly suggests that the original tumor was of lobular histology. It validates the congruence analysis above that the PDO and three PDXs (PDX.0, PDX.1A and PDX.1B) are representative of the ILC tumor cohort compared to IDC.

We then applied the default pathway selection criteria, adequate size (30< size <200) and |$NES$|>1.5 and selected 14 pathways as in the first cell line case study. As rationalized before, we again manually included the KEGG Cell Adhesion Molecules in addition to the 14 pathways. The pathway-specific congruence heatmap revealed performance of the six cancer models originating from the same patient (171881–019-R) (Fig 6B). Variations in performances were seen in the models even though they were developed from the same patient. Of the four pre-selected cancer models, PDO.1 had the largest pathway deviance score ($DS_{path}$ = 0.455) while PDX.1B has the smallest ($DS_{path}$ = 0.294) in "KEGG Cell Adhesion Molecules" pathway (Fig 6B) although none of them was statistically significant in lack of congruence. We next investigated KEGG topological network plot (S8 Fig) for the "KEGG Cell Adhesion Molecules" pathway comparing PDO.1 and PDX.1B. The expression of *CADM*1, *CADM*3, *CDH*2 was down-regulated ($DS_{gene}$ < −1.5) while expression of three other genes (*CDH*15, *NRXN*2, *L1CAM*) was upregulated ($DS_{gene}$ > 1.5) in PDO.1 (S8(B) Fig), compared with average expression of ILC tumor, while we observed better congruence in PDX.1B (S8(A) Fig). The comparison between PDX.1B and PDO in violin plots, with the gene expression distribution in IDC and ILC tumors as reference, verified that PDX.1B was a better model for ILC (Figs 6C and S7).

**Table 2. SDA-based genome-wide congruence summary for six models from patient 171881–019-R.** Later passages of PDX models have worse congruence (i.e., larger deviance scores).

| Sample ID | Label name | Model type | Passage | $DS_{SDA}^{(ILC)}$ | $P_{SDA}^{(ILC)}$ | Identified as ILC |
|---|---|---|---|---|---|---|
| APW-DS2 | PDX.0 | PDX | 0 | 0.12 | 1.00 | Yes |
| APYF68 | PDX.1A | PDX | 1 | 0.56 | 0.99 | Yes |
| APWG05 | PDX.1B | PDX | 1 | 1.15 | 0.90 | Yes |
| APWG05PF7 | PDX.2A | PDX | 2 | 1.93 | 0.34 | No |
| APVG40_RG-G15 | PDX.2B | PDX | 2 | 3.56 | 0.00 | No |
| V1-organoid | PDO | PDO | | 1.73 | 0.51 | Yes |

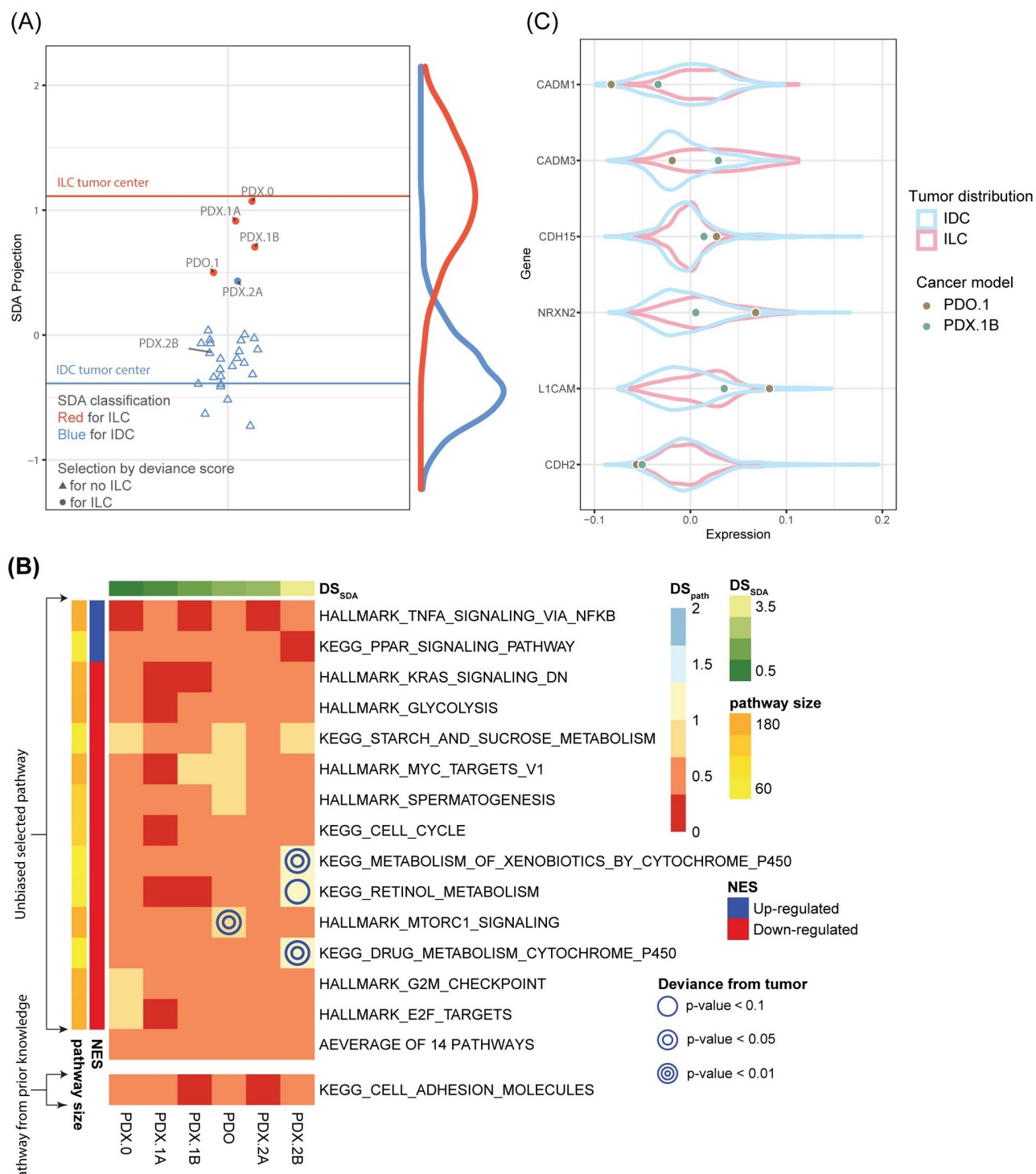

**Fig 6. Selecting representative PDO/PDX for ILC.** (A) SDA projected positions for PDO and PDX models from PDMR. Four models (three PDXs and one PDO; red circles) from the same patient (171881–019-R) were identified as candidate ILC models. Six models from this patient are labeled with the sample ID. High consistency was observed between SDA deviance scores and passages among PDX models. (B) Six models originated from the same patient were used for pathway-specific analysis. Six models show high congruence in the majority of 14 pathways and the Cell Adhesion pathway. (C) Violin plot shows the position of PDO.1 and PDX.1B on the six genes on which PDO.1 is discordant with.

## Discussion

Cancer models play a crucial role in cancer research for understanding carcinogenesis and drug development. However, how to best select the most congruent cancer model to faithfully represent a specific tumor subtype remains mostly unsolved, which is an urgent gap to fill given the increasing number of cell lines and PDOs being generated. In contrast to pure machine-learning-based methods in the literature, we developed a pipeline, CASCAM, to progressively select the most representative cancer model(s) by genome-wide pre-selection and pathway-specific mechanistic investigation using transcriptomics data. First, tumor and cancer model data are harmonized by Celligner (Module 1). The congruence evaluation combines merits of both machine learning and correlation-based approaches to pre-select cancer models (Module 2). In-depth bioinformatic tools provide iterative exploration of the most and least mimicked biological mechanisms of selected cancer models (Module 3).

The first example of this framework used ILC breast cancer data to select the most representative cell line(s), and it is demonstrated that CASCAM is suitable either in a supervised manner with prior knowledge of disease mechanism (e.g., cell adhesion pathway in ILC) or drug targeted pathways, or in an unsupervised manner when no prior knowledge is given. 14 cell lines were credentialed as ILC cell lines on the genome-wide evaluation by Module 2, and 10 of them (including user-specified MDA-MB-134VI) were used for pathway-specific analysis in Module 3. Though widely used in ILC research [32, 33], MDA-MB-134VI was not congruent with ILC tumors on the genome wide and in the "KEGG cell adhesion molecules" pathway. All results combined together indicated that CAMA1, UACC3133, SUM44PE, HCC2218, and IPH926 were recommended in order as appropriate cell lines for ILC research.

DU4475 is an example of the necessity of pathway-specific analysis (Module 3). As this cell line is E-cadherin positive, estrogen receptor positive [59], and without $CDH1$ mutation detected [60], DU4475 does not exhibit features of the classic ILC subtype. However, as epithelial-mesenchymal transition (EMT) preferentially occurs in basal cell lines [61], it often accounts for reduced $CDH1$ and $CDH2$ expression, which is also the key features of ILC tumors. Therefore, DU4475 was genome-wide classified as ILC. Importantly, pathway-specific analysis provided higher resolution to differentiate IDC and ILC, with DU4475 being dissimilar to ILC on average of the 14 selected pathways ($DS_{path} = 0.734$, $pval = 0.093$) and in the "KEGG cell adhesion molecules" pathway ($DS_{path} = 0.917$, $pval = 0.026$) and finally was not selected as a representative ILC cell line by CASCAM.

In practice, researchers tend to credential cell lines according to the annotation of their origin or pre-specified mutations (e.g., $CDH1$ for ILC) if available. However, the origins might be mislabeled, and the cell line evolution in culture has uncovered the possibility of genetic diversification, weakening the credibility of the original annotation. On the other hand, selected mutations cannot guarantee eligibility for a cell line. In our study; for example, we observed large genomic differences between SUM44PE ($P_{SDA} = 0.987$, $DS_{SDA} = 0.024$) and 600MPE ($P_{SDA} = 0.125$, $DS_{SDA} = 2.013$) although both have $CDH1$ mutation and are ER+, which are essential features of ILC. Therefore, the proposed CASCAM captures systems information in pathways, topological networks and genes and provides a thorough congruent investigation of the cell lines.

We also extended our framework to examine congruence of PDO and PDX cancer models to ILC tumors in the second case study. Of 11 PDOs and 136 PDXs in the PDMR database, only four from the same patient were credentialed as ILC in Module 2 evaluation. Strikingly, this tumor and model while not annotated as ILC has a $CDH1$ mutation, suggesting that CASCAM authenticated a new model of ILC. Aside from offering a "yes" or "no" answer, CASCAM can score the cancer models according to how similar they are to the targeted tumor

cohort. We therefore observed a progressive deviation trend for PDX samples over passages, which is consistent with recent reports that PDX often undergo murine-specific tumor evolution and congruence decays over passages [62, 63]. In fact, due to discrepancies in drug response for late-passage PDXs, recent studies have suggested design to use early-passage PDX models [64]. In addition, our result shows inconclusive comparative performance of PDO and PDX, which is consistent with current reports in the literature. PDOs offer advantages such as being an in vitro system with lower costs yet maintaining cellular heterogeneity [65, 66]. However, they lack of immune system components and loss of stromal components over time represent limitations in representing the tumor microenvironment [67]. On the other hand, PDXs preserve the original tumor architecture and include stromal and at least some immune components [68, 69], but again, PDX development and maintenance is time and resource intensive.

The current CASCAM has limitations and multiple directions of development are ongoing. The methodologies are now developed for transcriptomic data evaluation. As multi-level omics data (e.g., mutation, copy number variation, methylation and miRNA expression) are becoming affordable and prevalent, an extended congruence framework for evaluating cancer models with multi-omics data will provide deeper insight. Secondly, congruence analysis using single cell RNA-seq or single cell multi-omics data will provide a high-resolution understanding of clonal and micro-environment information for selecting the most representative cancer model, which is also an on-going work. Thirdly, the current framework is built upon binary contrast (i.e., ILC versus IDC). An extension to evaluating multi-class (i.e., three or more tumor subtypes) scenario is also a future direction. Currently, the molecular congruence we focus is on transcriptomic resemblance in genome-wide, pathway or gene level. The method can be extended to incorporate additional information, such as drug response, when available. Finally, the goal of CASCAM is to identify the most congruent cancer model from a long list of candidates to represent a target tumor cohort. For precision medicine, one may be interested in quantifying congruence of a PDO compared to the tumor from the patient origin. CASCAM can be easily extended for that purpose.

Collectively, we demonstrated CASCAM as a comprehensive and effective congruence evaluation tool for selecting the most representative cancer model for investigating cancer pathways and ultimately for precision medicine. An R package, CASCAM, with an interactive app is publicly available (https://github.com/jianzou75/CASCAM) to facilitate the use of our proposed framework.

## Materials and methods

### Gene expression data

Gene expression matrices in raw read count and transcripts per million (TPM) versions for 9,264 The Cancer Genome Atlas (TCGA) pan-cancer tumor samples were downloaded from Gene Expression Omnibus (https://www.ncbi.nlm.nih.gov/geo) with query ID GSM1536837 [46], and there were 960 breast cancer primary tumor samples with histology annotated as IDC or ILC. Log2-TPM gene expression data for 1,248 Cancer Cell Line Encyclopedia (CCLE) pan-cancer cell line samples were taken from DepMap Public 19Q4 file [70] (https://depmap.org/portal/ccle), and there were 65 breast cancer cell lines. Due to the limited representation of ILC cell lines in the CCLE project, we further included seventeen cell lines from an ongoing project (R01CA252378), namely Invasive Lobular Cancer Cell Line Encyclopedia (ICLE). The following eight cell lines were overlapping in ICLE and CCLE datasets: CAMA1, HCC1187, HCC2218, MDA-MB-134, MDA-MB-453, MDA-MB-468, SKBR3, and ZR7530. Those from ICLE were annotated as I, those from CCLE (sequencing data from Sequence

Read Archive (SRA) under accession number PRJNA523380) were annotated as C, and the processed CCLE data directly from DepMap were not annotated. Gene expression data of the breast cancer PDO and PDX models in TPM were obtained from NCI Patient-Derived Models Repository (PDMR) database (https://pdmr.cancer.gov/), and was log transformed for downstream evaluation. The genetic variants (e.g. mutations) in PDMR was extracted from whole genome sequence and annotated through oncoKB annotation pipeline version 1.1.0 [71].

## Gene expression normalization between tumor and cell lines

The gene expression matrices from tumors and cell lines are not directly comparable. We evaluated three different approaches for normalization. Quantile normalization is a widely used method to achieve equal quantiles across all the samples ("normalize.quantiles" function in preprocessCore [72] package). ComBat [21] is method for batch effect correction under empirical Bayes frameworks, where we treated tumor and cell lines as two different batches ("ComBat" function in sva [73] package). Celligner is a two-step machine learning method specifically developed for tumor and cell line normalization. The first step is to remove systemic differences, such as normal cell contamination, between tumor and cell lines using contrastive principal component analysis (cPCA). The second step is to perform further normalization using mutual nearest neighbors (MNN) [74]. We used the default parameters in Celligner implementation (celligner package), using either 960 breast cancer tumor samples or all 9,264 pan-cancer samples to harmonize the datasets.

## Differential expression analysis and gene set enrichment analysis

We applied DESeq2 [75] R package using TCGA tumor read count data for differential expression analysis (IDC vs. ILC). A gene with absolute fold change >1.5 and two-sided Benjamini-Hochberg adjusted p-value [76] <0.05 was defined as "differentially expressed (DE)". For gene set enrichment analysis, we used fgsea R package. Kyoto Encyclopedia of Genes and Genomes (KEGG) and Hallmark gene sets in the Molecular Signatures Database (MSigDB) were analyzed, and log2 fold changes from the differential expression analysis were used for gene ranking.

## Machine learning methods

We compared 16 machine learning methods, including sparse discriminant analysis (SDA), random forest on pre-filtered transformed data* (CancerCellNet), robust sparse discriminant analysis (RSDA), logistic regression with elastic net (ElasticNet), logistic regression with ridge penalty* (RidgeRegress), K nearest neighbors (KNN), majority voting according to 25 highest Pearson correlated tumor samples* (Pearson25), linear discriminant analysis (LDA), random forest* (RandomForest), nearest template prediction* (NTP), subtype assignment according to the median of within subtype Spearman correlations* (SpearmanMed), subtype assignment according to the median of within subtype Pearson correlations* (PearsonMed), logistic regression (Logistic), and three convolutional neural networks which were originally optimized on pan-cancer datasets (1D-CNN, 2D-Vanilla-CNN, and 2D-Hybrid-CNN) [25]. Six of these methods (marked with asterisk in Table 1) have been extended and used in publications for cancer model prediction analysis. The following three prediction evaluations were performed: (1) Five-fold cross-validation on breast cancer histology (769 IDC vs. 191 ILC) using 960 TCGA BC samples. (2) Construction of prediction model on Celligner aligned TCGA BC samples (training set, 712 ER+ and 205 ER-) and validated on Celligner aligned CCLE BC cell line samples (testing set, 19 ER+ and 37 ER- cell lines). (3) Construction of prediction model

on Celligner normalized TCGA pan-cancer samples (training set, 960 BC and 960 non-BC) and validated on Celligner normalized CCLE cell line samples (testing set, 56 BC and 56 non-BC cell lines). To avoid accuracy calculation issue of imbalanced sample sizes, 960 TCGA non-BC tumor samples and 56 CCLE non-BC cell lines were randomly subsampled from 8,304 TCGA pan-cancer non-BC samples and 1,192 CCLE pan-cancer non-BC cell lines.

## SDA projected deviance score

The SDA projected deviance score, $DS_{SDA}$, was designed based on the sparse discriminant analysis (SDA) method [77] to quantify genome-wide dissimilarity between a cancer model and the targeted tumor subtype. We denote the tumor gene expression $N \times G$ matrix as X with N samples and G DE genes, the $N \times 2$ class indicator matrix as Y with $Y_{ik} = 1_{(i \in Ck)}$ for tumor sample $i$ belonging to targeted tumor subtype, and the cancer model gene expression as C. SDA extends linear discriminant analysis with elastic net to identify $(\boldsymbol{\theta}, \boldsymbol{\beta})$ by

$$\text{minimize}_{\boldsymbol{\beta},\boldsymbol{\theta}} \left\{ \| \ \boldsymbol{Y\theta} - \boldsymbol{X\beta} \ \|^2 + \gamma \boldsymbol{\beta}^T \boldsymbol{I\beta} + \lambda \| \ \boldsymbol{\beta} \ \|_1 \right\}$$

$$\text{subject to } \frac{1}{n}\boldsymbol{\theta}^T \boldsymbol{Y}^T \boldsymbol{Y\theta} = 1$$

where $\boldsymbol{\theta}$ is the optimal scores, $\boldsymbol{\beta}$ is the sparse discriminant vector, $\boldsymbol{I}$ is the identity matrix, $\gamma$ and $\lambda$ are nonnegative tuning parameters selected by cross-validation. An iterative algorithm is applied to solve the pair $(\boldsymbol{\theta}, \boldsymbol{\beta})$. The tumor and cancer model gene expressions are then projected to the direction of estimated $\boldsymbol{\beta}$–$\boldsymbol{X\beta}$ and $\boldsymbol{C\beta}$. The assignment probability, $P_{SDA}$, was calculated from the standard LDA on the reduced data matrix $\boldsymbol{X\beta}$ and $\boldsymbol{C\beta}$ in selected gene features.

To simplify annotation, we use $c_i$ to denote the projected value for cancer model $i$ and $\boldsymbol{t_k}$ to denote the projected tumor sample vector for subtype $k$. The SDA projected deviance score for cancer model $i$ in class $k$, is defined as $DS_{SDA}^{(i,k)} = |c_i - \hat{\mu}_k|/\hat{s}$ where $\hat{\mu}_k = median_k(\boldsymbol{t_k})$, $\hat{s} = mad_k(\boldsymbol{t_k} - \hat{\mu}_k)$, and $mad$ is abbreviation for scaled median absolute deviation. Intuitively, $\hat{\mu}_k$ and $\hat{s}$ are robust forms of mean and standard deviation, and $DS_{SDA}$ can be seen as a robust form of absolute t-statistics as the standardized distance of the cancer model to the center of tumor cohort on the SDA projected space. Smaller deviance score indicates higher congruence of the cancer model to the desired tumor subtype cohort. By setting the null hypothesis as $c_i = \mu_k$, the p-value of $DS_{SDA}^{(i,k)}$, denoted as $pval(DS_{SDA}^{(i,k)})$, is obtained from the distribution of tumor $\boldsymbol{t_k} \sim N(\mu_k, \sigma)$, where $(\mu_k, \sigma)$ are estimated by $(\hat{\mu}_k, \hat{\sigma})$. The ordinary bootstrap [78] with 1,000 times on the tumor projected data is performed to obtain the 95% confidence interval of $DS_{SDA}^{(i,k)}$ on the log2 scale. The implementation of this method is based on sparseLDA [77], caret [79] and boot [80] R package.

## Gene and pathway specific deviance score

We denoted the Celligner aligned gene expression for cancer model $i$ and gene $g$ as $c_{g,i}$ and for tumor samples in subtype $k$ and gene $g$ as $t_{g,k}$. Similar to SDA-projected deviance score, we defined the gene specific deviance score ($DS_{gene}$) for model $i$ and subtype $k$ in gene $g$ as $DS_{gene}^{(g,i,k)} = (c_{g,i} - \hat{\mu}_{g,k}/\hat{\sigma}_g$, where $\hat{\mu}_{g,k} = median_k(\boldsymbol{t_{g,k}})$ and $\hat{\sigma}_g = mad_k(\boldsymbol{t_{g,k}})$. The pathway specific deviance score ($DS_{path}$) for pathway $p$, cancer model $i$, and tumor subtype k is then defined, based on the $DS_{gene}$, as $DS_{path}^{(p,i,k)} = \text{geometric mean}_{g \in P^{(DE)}}(|DS_{gene}^{(g,i,k)}|)$, where $P^{(DE)}$ is the set of DE genes in pathway $p$. The geometric mean is proposed to reduce the effects of outliers. The significance levels of $DS_{path}$ (one-sided p-values) were defined similar to $pval(DS_{SDA})$, which were obtained from the null distribution empirically constructed by $DS_{path}$ of the tumor samples.

## Supporting information

**S1 Table. Summary of relevant publications.**
(DOCX)

**S2 Table. Celligner aligned data for 38 candidate cell lines and SDA projection weights.**
(XLSX)

**S3 Table. Summary table of the 38 candidate cell lines.**
(DOCX)

**S4 Table. Summary table of the pathway specific analysis for 9 unbiased-selected + 1 manually-included cell lines and 14 unbiased-selected +1 manually-included pathways.**
(DOCX)

**S5 Table. Confusion matrix of genome-wide predication for HER2+, ER+/PR+, and TNBC prediction.**
(DOCX)

**S6 Table. Summary table of the 11 PDO and 136 PDX BC models.**
(DOCX)

**S1 Fig. Assignment probabilities of three different methods.** (A) LDA; (B) SDA; (C) RSLDA.
(TIF)

**S2 Fig. Heatmap of pathway-specific deviance scores ($DS_{path}$) with 53 pathways (rows) and 9 genome-wide pre-selected cell lines +1 manually selected cell line (MDA-MB-134VI) (columns).** The genome-wide SDA projected deviance score ($DS_{SDA}$) is shown on the top sidebar and the pathway size and normalized enrichment score (NES) are on the left.
(TIF)

**S3 Fig. Violin plot for CAMA1 and CDK4 in KEGG Cell Adhesion Molecules.**
(TIF)

**S4 Fig. Cross-cancer validation of CASCAM.** (A) Genome-wide preselection identifies 5 out of 24 liver cell lines as ILC. (B) Pathway-based heatmap reveals that these 5 preselected cell lines significantly diverge from the ILC tumor center in terms of differentially expressed genes and associated pathways.
(TIF)

**S5 Fig. Selection of representative cell lines for TNBC.** (A) SDA projected scatterplot shows the position of each candidate cell line and 12 cell lines were selected for down-stream analysis. (B) heatmap for the comparison of cell lines in KEGG WNT signaling pathway (BT549 and HCC1599 highlighted).
(TIF)

**S6 Fig. UMAP of Celligner alignment between tumors and PDX/PDO models.** (A) Three distinct clusters were observed. The small cluster on the left consists of a seemingly rare breast cancer subtype, the upper-right cluster includes mostly non-basal samples, and the lower-right cluster includes mostly basal samples. (B) UMAP is redrawn when the small cluster in (A) is removed.
(TIF)

**S7 Fig. Violin plot for PDO.1 and PDX.1B in KEGG Cell Adhesion Molecules.**
(TIF)

**S8 Fig. Topological plots for APWG05PF7 (PDX.1B) and V1-organoid (PDO) from the same patient (171881–09-R) in KEGG Cell Adhesion Molecules.** (A) APWG05 (PDX.1B). (B) V1-organoid (PDO.1).
(TIF)

**S9 Fig. Scatterplot for relation between $DS_{SDA}$ and $DS_{path}$ in PDO and PDX models across (A) fibroblast and immune pathway, (B) metabolism of xenobiotics by cytochrome P450 (KEGG ID: hsa00980) and (C) cell death (MSigDB ID: M5902).**
(TIF)

## Author Contributions

**Conceptualization:** Jian Zou, Tianzhou Ma, George C. Tseng.

**Data curation:** Jian Zou, Osama Shah, Jennifer M. Atkinson.

**Formal analysis:** Jian Zou, Yu-Chiao Chiu, George C. Tseng.

**Funding acquisition:** Steffi Oesterreich, Adrian V. Lee, George C. Tseng.

**Investigation:** Jian Zou, George C. Tseng.

**Methodology:** Jian Zou, George C. Tseng.

**Resources:** Osama Shah, Steffi Oesterreich.

**Software:** Jian Zou.

**Supervision:** Steffi Oesterreich, Adrian V. Lee, George C. Tseng.

**Visualization:** Jian Zou, George C. Tseng.

**Writing – original draft:** Jian Zou, George C. Tseng.

**Writing – review & editing:** Jian Zou, Osama Shah, Yu-Chiao Chiu, Tianzhou Ma, Jennifer M. Atkinson, Steffi Oesterreich, Adrian V. Lee, George C. Tseng.

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
