## [Decision Letter · Decision Letter 0]

28 Aug 2023

Dear Prof. Tseng,

Thank you very much for submitting your manuscript "Systems approach for congruence and selection of cancer models towards precision medicine" for consideration at PLOS Computational Biology.

As with all papers reviewed by the journal, your manuscript was reviewed by members of the editorial board and by several independent reviewers. In light of the reviews (below this email), we would like to invite the resubmission of a significantly-revised version that takes into account the reviewers' comments.

We cannot make any decision about publication until we have seen the revised manuscript and your response to the reviewers' comments. Your revised manuscript is also likely to be sent to reviewers for further evaluation.

Sincerely,

Pedro Mendes, PhD

Section Editor

PLOS Computational Biology

Reviewer's Responses to Questions

**Comments to the Authors:**

Reviewer #1: Zou et al have provided a useful, clearly written manuscript describing the method CASCAM. CASCAM builds off of the expression alignment tool Celligner to include machine learning methods for cancer cell type classification with an interpretability module. These additions provide utility that will benefit the research community, and the authors have provided an accessible GitHub implementation of the software.

Overall, I have a positive assessment of the work, with two relatively minor comments.

Line 306: “Fig. 6A showed a clear pattern that PDX.0 has almost perfect DSSDA congruence to represent ILC but the deviance score increased with increasing passage numbers (also see Table 2 Column 5), indicating that the xenografts may evolve and be affected by the microenvironments in mice and deviate from the original tumor over time.“

A common difference between later passage xenografts and early passages is that stromal cells such as fibroblasts and immune cells do not persist. The authors’ statement suggests, on the other hand, that they believe that the cancer cells evolve between early and late passages. The authors should clarify this point by doing a more thorough pathway analysis using their module 3 to determine whether the difference in assignment of ILC status is related to fibroblast and immune pathways. To clarify why the PDO is identified as ILC, it would be useful for the authors to determine from PDMR whether the PDO was derived from the ILC-classified PDX lineages, the non-ILC PDX lineages, or through some other lineage not downstream of the observed PDX models.

Line 389: “PDO is widely believed to be a highly conserved cancer model promising for precision medicine development and superior to PDX [7,50].”

This statement is not accurate. Although PDOs have promising uses, there is not a consensus that they are superior to PDXs. For example, PDX drug treatment experiments remain the standard used by the NCI for deciding on advancement to human clinical trials. Please adjust your description.

Reviewer #2: This manuscript is focused on the analysis of breast cancer cell lines, PDOs, PDXs and breast cancer samples. They have proposed a three module analysis system. Starting with the usual issue of batch correction and normalization they have integrated various datasets, demonstrating that newer methods offer advances over some well used modules. The second module uses a machine learning approach to assign samples and the third module offers information about pathways. This was applied to breast cancer cell lines and PDOs in two case studies, both being focused on ILC. The manuscript is well written, the CASCAM package is available in R and it is a valuable resource to those without established analysis pipelines in their lab.

The major limitation is that the analysis and proof of concept work was limited to ILC samples. This (as they state) only composes 10-15% of breast cancer. As such, this does NOT adequately profile the CASCAM package for general use. In order for the manuscript to be widely read and applied, additional analysis should be completed for other major breast cancer subtypes. At a minimum I would request that HER2+ve, ER+/PR+ and TNBC experiments be completed and then added to the figures that were provided. Given that the R package is complete, this request does not seem excessive and would provide a much more interesting and important validation.

Other concerns are all minor, from the use of contractions in the manuscript to Figure 5 Panel C being overly pixelated.

Reviewer #3: Zou et al. described a computational approach, CASCAM, to determine how well currently available cancer models recapitulate the biology of cancers from patients. Selection of the relevant model systems is crucial to provide mechanistic understanding of the disease as well as identifying new targatable vulnerabilities, not to mention their use for drug screening purposes. The authors show that CASCAM can rank cancer model systems based on genome-wide scale transcriptional profiles and also based on pathway and gene-level information, determining the cancer models with higher congruence/similarity relative to tumor types for which transcriptional profiles were available. Although the algorithm does identify cancer models that are potentially more appropriate for their cancer of interest, invasive lobular breast carcinoma, the work lacks computational novelty as it mostly uses already curated datasets, as well as traditional machine learning algorithms for cancer classification.

Major comments:

1) Virtually all claims made by authors, that CASCAM is unique with regards cancer model classification and prioritization, could also be performed with currently available methods based on downstream analysis. For instance, by raking cancer models using classification scores from current methods, users could then select the models with higher classification probability and then, identify pathways and genes conserved or divergent between the ranked models and their in vivo tumor counterparts. The authors simply implemented a downstream metric based on the sparse linear discriminant analysis (SDA) projected space that would allow them to evaluate some of the cancer model heterogeneity relative to the bulk tumor measurements.

2) The authors also claim that their machine learning model, based on SDA provides an interpretable machine-learning model because gene selection leads to gene signatures that make the model interpretable. This is not entirely true as far as I can tell, the model itself does not provide information for which features/genes are actually important to drive the classification scores. Several machine learning methods use gene signatures as features for training. This does not make such models fully interpretable.

3) CASCAM lacks computational novelty, it simply combines already available methodologies (Cellaligner, SDA, pathway analysis) into a workflow for cancer model prioritization. In the lack of experimental follow-up and in absence of clear biological insights related to the biology of their tumor of interest, it feels like CASCAM does not represent a significant advance over prior methods on the same task, especially because there were no additional comparisons on the molecule profiles of cancer models identified by ElasticNet or 2D-Hybrid-CNN, etc.

4) Authors should also include a comparison of cancer models other than their cancer of interest, to determine whether the pathway and gene-level modules of CASCAM indeed capture biological processes specific to cancer models for ILC.

Minor comments:

1) Authors should cite Figure panels, not the approaches themselves (A-D), as they do in Figure 1.

2) It is not clear the rationale by which authors select pathways based on large (and not small) P-values. In my experience this is counterintuitive as authors seem to select cell lines that are not statistically significant. I understand what they mean, but this should be more clearly stated to avoid confusion. See lines 165-166, for instance.

3) Figure 5C does not add anything to the manuscript. The resolution is also pretty bad.

4) Authors should more clearly state their conclusions and better explain their results. In line 330, for instance, the authors describe the violin plot but an actual conclusion is not clearly stated

**Have the authors made all data and (if applicable) computational code underlying the findings in their manuscript fully available?**

Reviewer #1: Yes

Reviewer #2: Yes

Reviewer #3: Yes

PLOS authors have the option to publish the peer review history of their article (what does this mean?). If published, this will include your full peer review and any attached files.

Reviewer #1: No

Reviewer #2: No

Reviewer #3: No
---

## [Decision Letter · Decision Letter 1]

12 Dec 2023

Dear Prof. Tseng,

We are pleased to inform you that your manuscript 'Systems approach for congruence and selection of cancer models towards precision medicine' has been provisionally accepted for publication in PLOS Computational Biology.

Best regards,

Pedro Mendes

Section Editor

PLOS Computational Biology

Reviewer's Responses to Questions

**Comments to the Authors:**

Reviewer #1: The authors have addressed my concerns.

Reviewer #2: The authors appropriately responded to my major concern and have appropriately revised the manuscript.

Reviewer #3: The authors addressed all of my concerns and I don't have any further comments.

**Have the authors made all data and (if applicable) computational code underlying the findings in their manuscript fully available?**

Reviewer #1: Yes

Reviewer #2: Yes

Reviewer #3: Yes

PLOS authors have the option to publish the peer review history of their article (what does this mean?). If published, this will include your full peer review and any attached files.

Reviewer #1: No

Reviewer #2: No

Reviewer #3: No

---

## [Editor Report · Acceptance letter]

5 Jan 2024

PCOMPBIOL-D-23-00715R1 

Systems approach for congruence and selection of cancer models towards precision medicine

Dear Dr Tseng,

I am pleased to inform you that your manuscript has been formally accepted for publication in PLOS Computational Biology. Your manuscript is now with our production department and you will be notified of the publication date in due course.

With kind regards,

Timea Kemeri-Szekernyes
